# Comparative Gene Enrichment Analysis for Drought Tolerance in Contrasting Maize Genotype

**DOI:** 10.3390/genes14010031

**Published:** 2022-12-22

**Authors:** Syed Faheem Anjum Gillani, Adnan Rasheed, Asim Abbasi, Yasir Majeed, Musawer Abbas, Muhammad Umair Hassan, Sameer H. Qari, Najat Binothman, Najla Amin T. Al Kashgry, Majid Mahmood Tahir, Yunling Peng

**Affiliations:** 1Gansu Provincial Key Lab of Arid Land Crop Science, Lanzhou 730070, China; 2College of Agronomy, Hunan Agricultural University, Changsha 410128, China; 3Department of Environmental Sciences, Kohsar University, Murree 47150, Pakistan; 4Department of Agronomy, University of Agriculture Faisalabad, Faisalabad 38000, Pakistan; 5Research Center on Ecological Sciences, Jiangxi Agricultural University, Nanchang 330029, China; 6Genetics and Molecular Biology Central Laboratory, Department of Biology, Aljumum University College, Umm Al-Qura University, Makkah 24243, Saudi Arabia; 7Department of Chemistry, College of Sciences and Arts, King Abdulaziz University, Rabigh 21911, Saudi Arabia; 8Department of Biology, College of Science, Taif University, P.O. Box 11099, Taif 21944, Saudi Arabia; 9Department of Soil and Environmental Sciences, Faculty of Agriculture, University of Poonch, Rawalakot 12350, Pakistan

**Keywords:** GO analysis, maize, GSEA, SNP, PCR, drought tolerant

## Abstract

Drought stress is a significant abiotic factor influencing maize growth and development. Understanding the molecular mechanism of drought tolerance is critical to develop the drought tolerant genotype. The identification of the stress responsive gene is the first step to developing a drought tolerant genotype. The aim of the current research was to pinpoint the genes that are essential for conserved samples in maize drought tolerance. In the current study, inbred lines of maize, 478 and H21, a drought-tolerant and susceptible line, were cultivated in the field and various treatments were applied. The circumstances during the vegetative stage (severe drought, moderate drought and well-watered environments) and RNA sequencing were used to look into their origins. In 478, 68%, 48% and 32% of drought-responsive genes (DRGs) were found, with 63% of DRGs in moderate drought and severe drought conditions in H21, respectively. Gene ontology (GO) keywords were explicitly enriched in the DRGs of H21, which were considerably over-represented in the two lines. According to the results of the GSEA, “phenylpropanoid biosynthesis” was exclusively enriched in H21, but “starch and sucrose metabolism” and “plant hormone signal transduction” were enhanced in both of the two lines. Further investigation found that the various expression patterns of genes linked to the trehalose biosynthesis pathway, reactive oxygen scavenging, and transcription factors, may have a role in maize’s ability to withstand drought. Our findings illuminate the molecular ways that respond to lack and offer gene resources for maize drought resistance. Similarly, SNP and correlation analysis gave us noticeable results that urged us to do the same kind of analysis on other crops. Additionally, we isolated particular transcription factors that could control the expression of genes associated to photosynthesis and leaf senescence. According to our findings, a key factor in tolerance is the equilibrium between the induction of leaf senescence and the preservation of photosynthesis under drought.

## 1. Introduction

Maize (*Zeal mays* L.) is a significant cereal crop. Maize production around the world is adversely impacted by drought stress. The breeding of drought-tolerant genotypes is a practical way of addresing the water shortage problem. This calls for a thorough understanding of the sub-atomic elements involved in the reactions caused by drought pressure in maize, as well as genomic means such as genomic determination, hereditary designing, and genomic alteration for drought tolerant improvement. Plants’ response to stress can be categorized into four stages: alert stage, aggregation stage, support stage, and fatigue stage. The sub-atomic reactions to the drought pressure of plants include more administrative and useful assets [1].

The transcription factor (TFs) is a significant controller protein that can unequivocally direct initiation and the restraint of target qualities. Then, plants reduce water loss as a result of partially closing stomata, effectively taking up water from underground by root foundation, and regulating metabolic cycles to correspond with existing carbon reserve. With the increment of drought, osmolytes, such as proline and solvent sugars, would aggregate to keep up with the cell turgor pressure. Moreover, various antioxidants, for example, superoxide dismutase (SOD, catalyze), reduce the toxic effect activity in reactive oxygen species (ROS) [2].

The stacking bearing connections inside the receiver wall organization, for example, caused specific revision under the activity of the cell wall’s changing proteins, and the guideline of the phenylpropanoid pathway prompted a high collection of lignins in the cell wall. Quality delivery network policies play an essential role in plant drought reactions. Various transcriptions are responsible in various crops. In two wheat genotypes, different responses to drought stress were found bases on transcription analysis. [3].

The results showed that in wheat the auxin and abscisic acid (ABA) receptor related to ROS and the biosynthesis of the cell wall was up regulated, while ethylene receptors were down regulated. In maize, the moderate response to drought were linked to DNA and cell cycle. It was shown that qualities engaged with metabolic pathways and the biosynthesis of optional digestion showed a reaction to lack, and the statement of a few qualities connected with ABA signal transduction were prompted under drought [4]. The plant roots detect changes in water and soil oxygen content. It was shown that exceptional drought-responsive components were associated with various parts of the root zone in maize. The qualities connected with ROS and carbon digestion, and qualities linked to layer carrier, were essentially improved in drought-responsive qualities of 0–3 mm root peak, with 3–7 mm upon root chimp, separately. Comparable outcomes were likewise acquired in another research study utilizing two deep-rooted maize lines. Cell reinforcement chemical qualities were demonstrated to assume the essential parts in drought tolerance in the root transcriptomes analysis of essential root under drought stress, conducted by RNA sequencing, and the results demonstrated the expression of genes under low water stress. A few members from the MAPKKK family enhanced drought tolerance in leaf and maize stems, recommending the significance of the MAPKKK family under drought stress [5].

Plant drought resilience is generally assessed by variances in aggregates and physiological and bio-chemical reactions. Drought plants can keep up with constant morphological features throughout delayed times of drought pressure by maintaining water-holding capability. According to one perspective, drought-resistant lines have some development to do to increase their capacity to keep moisture content in their root system. The entire root development of maize may be made to conduct less research on the soil profile and extend into more soil by reducing the number of particular cortical cells in maize, while simultaneously improving their size. This allows maize to uptake greater water from the soil. The thickness of the leaves may be reduced, and the opening of the stomata in the plants can be regulated to increase the plant’s resistance to drought [6].

Furthermore, during conditions of drought, plants are able to maintain the homeostasis of their cells via the process of osmoregulation, which is orchestrated by soluble sugars. This helps to decrease the amount of water that is likely to be present in the cells. Water stress also mandates the peroxide-rummaging catalyst structure to eradicate the quantity of responsive oxygen species incited by drought. This is necessary since ROS may cause damage to the cell film structures, which can ultimately lead to cell death. The activity of the cell reinforcement (nonenzymatic and enzymatic) framework addresses an important file for evaluating maize’s resistance to drought. Biochemical and physiological alterations between deep-rooted lines and transgenic lines have allowed for the identification of drought-tolerant lines. Despite this, an exclusive explanation for maize’s drought resistance could not be provided by these advances, since different types had different genetic basis [7].

It is realistic that the drought reaction of plants includes a chain of reactions. In this approach, it is planned that reading the article would help agricultural plants become more resilient to drought by providing a realistic illustration of these features. To date a couple of droughts, and tolerant features, for example, H21 and 478 in maize, were distinguished by conventional sequencing strategies and practically depicted. Many key qualities were suggested in drought-tolerant maize; however, an overall significant stretch was expected to explain the elements of these drought reaction qualities. RNA-seq is an incredible technique for the enormous scope ID of drought-responsive attributes with minimal expense, high throughput, and high responsiveness. It can work with mining crucial drought-tolerant qualities in plants, e.g., maize [8].

For instance, research findings have demonstrated that the overexpression of cell wall oxidative metabolism characteristics makes it possible for maize recombinant inculcated lines to procure water stress flexibility under water shortages; the attributes associated with cell wall reformation are involved with water stress chemical reactions in a drought resistant corn line; and the compositions of proteins and carbohydrates are associated with water stress support. Through the process of lighting seeds from an intrinsic maize line, we were able to create a deficit resistant line for this study. To investigate the components of drought opposition of H21, we have analyzed physiological, bio-chemical, and transcriptomic changes between H21 and 478 lines. Our outcomes would feature the drought-tolerant marks of H21 and add to the distinguishing resilient and useful investigations of novel drought resilience qualities in maize [9].

## 2. Materials and Methods

### 2.1. Plant Material and Experimental Design

Seeds of maize inbred lines, drought-tolerant (478) and drought-sensitive (H21) were given by College of Agronomy, Gansu Agricultural University, Lanzhou, China. Seeds were sown in pots holding standard preserving soil. The pots were kept in a climate that was sustained at a temperature of 25 ± 5 degrees Celsius, with a relative humidity of 65 ± 5 percent, and a light/dark cycle of 16/8 h. All the plants were adequately sprinkled to keep the moistness level persistent till 10 days after emergence; after that, drought treatments began. Treatments included two different degrees of drought conditions (no stress and stress), as well as three different levels of BR (0, 25, and 50 mL). To simulate the effects of drought, an application of fifty milliliters of 20 percent PEG 6000 every 3 days was made, whereas the control area received applications of pure water. For the transcriptome study, plants were chosen after they had reached one of four fully formed leaf stages. Samples were then kept at −80 degrees Celsius in liquid nitrogen for 7 days. The experiment was laid out in a completely randomized design (CRD), and there were three separate replications of the experiment.

### 2.2. RNA Extraction, Construction of Library, and Sequencing

Pure and simple RNA was detached using Trizol reagent (Thermo Fisher, Carlsbad, CA, USA, 15596018). Total RNA quantity and quality was determined using the Bioanalyzer 2100 and RNA 6000 Nano LabChip Kit (Agilent, Santa Clara, CA, USA, 5067-1511) and RNA assays from the first order with a RIN greater than 7, 0 was used in building sequencing libraries. After removal of the whole RNA, the mRNA residual in the purified RNA (5 μg) was cleaned by two cycles of filtration with Dynabeads Oligo (dT) (Thermo Fisher, Carlsbad, CA, USA). Once filtered, the mRNA was cleaved into shorter fractions using higher temperature divalent cations (using the NEB Magnesium RNA Fragmentation Module, cat. e6150, Ipswich, MA, USA) for 5–7 min at 94 °C. The cut RNA fragments were then reverse disassembled to generate cDNA using the Super ScriptTM II reverse transcriptase enzyme (Invitrogen, cat. 1896649, Carlsbad, CA, USA). The cDNA was then used to combine the second desert DNA named U with DNA polymerase I from *E. coli* (NEB, cat.m0209, USA) and RNase H (NEB, cat.m0297, USA) and dUTP solution (Thermo Fisher, Carlsbad, CA, USA, USA, ref. R0133). The crude ends of each thread receive an A-base coating, which prepares the fibers for tightening into the taped connectors.

Each particular connector was constructed with a T-base shade so that it could ligate to the A-followed segmented DNA and complete the connection. AM Pure XP dabs were used for the purpose of carrying out size verification after twofold document connections had been affixed to the various components. Following the power labile UD Genzyme (NEB, cat.m0280, USA) treatment of the U-named second-deserted DNAs, the ligated things were escalated with PCR using the following conditions: starting denaturation at 95 °C for 3 min; 8 examples of denaturation at 98 °C for 15 s, treating at 60 °C for 15 s, and increase at 72 °C for 30 s; and a short time later last extension at The expansion sizes of the cDNA libraries that came before this one varied, on average, from 300 to 50 base pairs in length. We finished the 2 × 150 bp matched end sequencing (PE150) utilizing an Illumina NovaseqTM 6000 after the shipper advised that we do so (LC-Bio Technology Co., Ltd., Hangzhou, China).

### 2.3. Sequence and Filtering of Clean Reads

A cDNA library was created by design and sequenced utilizing pooled RNA from experiments conducted using the Illumina Novaseq TM 6000 succession stage. We used the Illumina paired-end RNA-seq method to sequence the transcriptome; readings generated from the sequencing equipment comprise raw read out with devices or low-quality bases, which have an impact on the subsequent assembly and analysis (Martin, 2011). As a result, readings were further filtered using Cutadapt to provide high-quality clean reads. The following were the criteria: eliminating reads that contain adapters, poly A and poly, more than 5% of unknown nucleotides (N), low-quality reads that contain more than 20% of low-quality (Q-value 20) bases, and reads that contain adapters. Then, FastQC was used to assess the sequence’s quality. It was noted that the data’s Q20, Q30, and GC-content had not been tainted. Following that, readings with clean endings totaling Gigabase Pairs were produced [7]. The NCBI Gene Expression Omnibus (GEO) databases or the NCBI Short Read Archive (SRA) databases have received the raw sequencing data along with accession numbers. These two archives each have an accession number.

## 3. Methods

### 3.1. Differently Expressed Genes (DEGs)

Differentially expressed genes refer to the up-regulated genes and down-regulated genes between samples or after different treatments of the same sample. The genes are usually screened in terms of fold difference and significant level. The consistency fold FC ≥ 2 or FC ≥ 0.5 (that is, the absolute value of log 2 FC ≥ 1) and q value < 0.05 (|log 2 fc| ≥ 1 & q < 0.05) as the standard (multi-group comparison without differential fold, the genes with q < 0.05 were screened for genes with statistical differences among multiple groups), and the genes screened out were considered to be differentially expressed genes, and the significant column in the result file was displayed as “yes”. You can also change the above two conditional thresholds for screening differentially expressed genes to obtain differentially expressed genes that align with your experimental design (using the multi-column screening function of excel).

The input data for the differential expression analysis of genes (transcripts) is the reads count data of genes (transcripts). The *p*-value calculation model based on the negative binomial distribution is used to calculate the *p*-value. Repeat (comparison between samples) using Edger for analysis; multi-group comparison using edgeR for analysis; using BH to correct *p* value to obtain q value (FDR value, p.adj value); the fold difference is the average expression level of the experimental group The value was divided by the mean expression level of the control group [10]. The raw sequencing data were produced on the Illumina HiSeq 2000 (Illumina, San Diego, CA, USA).

### 3.2. GSEA Analysis

Enrichment analysis based on a hypergeometric test generally needs to use the information of differentially expressed genes, that is, a change threshold needs to be set to determine which genes are significantly different and which genes are not very different (general or common thresholds may not be applicable to all systems). Traditional enrichment analysis may yield few or no results when single gene changes are weak. Gene Set Enrichment Analysis (GSEA) may more thoroughly describe the protective mechanism of a system component and efficiently compensate for standard enrichment analysis’s inability to properly mine the information from minor genes, which can interact with traditional enrichment analysis. Replenish. Due to the different principles of formal enrichment analysis, the results of GSEA analysis are different from those of conventional enrichment analysis [11].

### 3.3. GO Analysis

Gene Ontology (GO) is an internationally consistent gene function cataloging system that runs vigorously reorganized standard vocabularies to systematically define the properties of genes and gene products in organisms. GO has three ontologies, which represent the molecular function (mf), cellular component (cc), and biological process (bp) of genes, respectively. The elementary component of GO is Word (entry, node, entry), and each term relates to a characteristic. Biological process: A biological process accomplished by various molecular activities, such as cell cycle, cell cycle arrest, cell cycle checkpoint, and positive execution of the execution phase of apoptosis. Processes such as positive regulation of the execution phase of apoptosis. Cell components: the location of the cell structure where the gene product performs its function, such as cytoplasm, nucleus, lysosome, and other structures. Molecular function: the activity of gene products at the molecular level, such as protein binding, ubiquitin-protein transferase activity, G protein-coupled receptor activity, etc. [12].

### 3.4. SNP Analysis

The full name of SNP is Single Nucleotide Polymorphisms, which refers to the genetic markers formed by the variation of a single nucleotide on the genome, with a large number and rich polymorphisms. Variations of single nucleotides in the genome include substitutions, transversions, deletions, and insertions. The diversity of single nucleotide base morphology can be divided into substitution (transition, CT, GA on its complementary strand) and transversion (transversion, CA, GT, CG, AT). SNPs with substitutional variants account for about 2/3, and the occurrence probability of other variants is similar. SNPs seem most commonly on CG sequences, most of which are C-to-T conversions, because C in CG is often methylated and becomes thymine after spontaneous deamination. A SNP is a single nucleotide variant with a frequency greater than 1% [13].

Depending on where the SNP is placed in the gene, it may be classified as either being in the non-coding area of the gene, the spacer portion of the gene (which is the portion between genes), or the coding section of the gene. These classifications can be found in the table below. Single nucleotide polymorphisms (SNPs) not in the coding regions of genes may still affect gene splicing, transcription factor binding, messenger RNA degradation, or RNA sequences in non-coding parts. The expression of a gene affected by this single nucleotide polymorphism (SNP) is called an expression of a single nucleotide polymorphism (ESNP). It can occur either upstream or downstream of the gene. The number of SNPs (coding SNPs, cSNPs) located in the coding region of genes is relatively small as, inside exons, the modification rate is only 1/5 of that of adjacent sequences. However, it is of great significance in studying hereditary diseases, so the study of cSNP is more concerned [14].

From the perception of the impact on the genetic traits of organisms, cSNP can be divided into two types: one is synonymous cSNP (synonymous cSNP), that is, the change of the coding sequence caused by the SNP does not affect the amino acid sequence of the protein it translates. The mutated base has the same meaning as the unmutated base. The other is non-synonymous cSNP (non-synonymous cSNP), which means that the base sequence can change the line of the protein translated based on it, thus affecting the protein function. Such changes are frequently the direct cause of changes in biological traits. About half of the cSNPs are non-synonymous cSNPs. InDel (insertion-deletion) refers to the insertion and deletion of small fragments in the sample relative to the reference genome, which may contain one or more bases. Based on the transcriptome level, SNP loci in coding regions were analyzed. According to the Hisat2 alignment results of each sample and the reference genome, each sample’s possible SNP and INDEL information was obtained by mpileup processing with Samtools software and then annotated with ANNOVAR [15].

### 3.5. Correlations Analysis

The association analysis of the gene expression information of the samples can better judge the clustering situation between the samples. The greater the correlation coefficient between the samples, the better the clustering of the samples. The Pearson correlation coefficient between samples and PCA (Principal Component Analysis) allows the repeatability between samples to be understood, which helps to exclude outlier samples. The abscissa and ordinate in the Pearson correlation coefficient graph are for each piece. The depth of color indicates the size of the correlation coefficient of the two samples. The closer to red (the coefficient is closer to 1) the higher the correlation; the closer to white the lower the correlation. Some sample types (such as clinical samples, animal models, tissue fluids, cell pools with low knockdown efficiency, etc.) may cause the correlation of samples within a group to be lower than those between groups due to heterogeneity [16].

### 3.6. QPCR Analysis

The Rotor-Gene 3000 detection equipment (Corbett Research, Singapore) was used to conduct the PCR experiments. One-Step PrimeScriptTM RT-PCR Kit (Perfect Real Time) was used (Takara, Shiga, Japan). The following ingredients were used in each set of reactions: 10 L of 2 One Step RT-PCR Buffer III, 0.4 L of each of the following: MCMVf (10 M), MCMVr (10 M), 0.8 L of probe (10 M), 0.4 L of EX TaqTM (Takara) HS (5 U/L), 0.4 L of PrimeScriptTM RT Enzyme Mix II, 1.0 L of total RNA or 1.0 L of The following amplification responses were carried out: forty cycles of 95 °C for 5 s, followed by 60 °C for 20 s. Six distinct reactions, including a water control, were used to assess the specificity of this TaqMan test. Under optimal reaction circumstances, the TaqMan probe was only able to identify strong fluorescent signals after reactions involving samples, whereas the indications after four more samples and the water control were superposed to the baseline. By contrasting the signals at various levels, these samples may be distinguished from the four other corn samples. Agarose gel electrophoresis was used to further evaluate the PCR results. While the four other assays did not show the unexpected weak band above the 67-bp band, the assessment with the taster did show the probable band of 67 bp.

### 3.7. Statistical Analysis

Statistical significances between drought treatment and control were tested using the—Newman–Keuls method at *p* < 0.05 by IBM^®^ SPSS^®^, 22 and data was generated by Sigma Plot 12.5.

## 4. Results

The results obtained after detailed analysis are as follows.

### 4.1. Differently Expressed Genes (DEGs)

Using the sequence map (SAM) files that were generated by Tophat2, aligned reads of the genes were computed in order to determine the genes that have differentially expressed proteins. Then, using the R program “DESeq2, differential gene expression analysis were carried out comparing the two sets of samples. Gene with |log 2 (fold changes)| > 1 and false discovery rate (FDR) < 0.02 were recognized as differentially expressed. To detect the DRGs, four comparison groups, i.e., LMC_LMD (Lv28 under moderate drought control vs. Lv28 under moderate drought), LSC_LSD (Lv28 under severe drought control vs. Lv28 under severe drought), HMC_HMD (H21 under moderate drought control vs. 478 under moderate drought), and HSC_HSD (H21 under severe drought control vs. 478 under severe drought) were included Figure 1. To identify genes responding to drought in the two lines, differential expression analysis was performed by comparing the gene expression profiles between different drought conditions.

In total, 100% and 51% of genes showed drought response under MD and SD in 478, respectively, among which 28% and 26% were up-regulated under MD and SD, while 16% and 12% were down-regulated, respectively, Figure 2. In the drought tolerant line H21, 68% and 3 3% genes showed response to drought stress under MD and SD, respectively, among which 62% and 59% genes were up-regulated, while 16% and 15% genes were down-regulated, respectively.

### 4.2. GO Analysis

The significance of GO enhancement analysis showed that GO terms were improved in the DRGs of the samples, as shown in Figure 3, with 15 in the metaphysics of “natural cycle”, nine in the cosmology of “sub-atomic capability”, and seven in the philosophy of “cell part”. There were many GO terms enhanced in the samples, including eight organic cycle terms, which were primarily about pressure reaction (e.g., “reaction to boost”, “reaction to stretch”, “reaction to abiotic improvement”, “reaction to biotic upgrade”, “reaction to endogenous improvement”, and “reaction to outside boost”). Be that as it may, these pressure reaction-related GO terms (with the exception of “reaction to boost”) were missing in H21 under MD. Likewise, “metabolic sugar cycle” and “extracellular district” were over-addressed in every one of the four DRG sets (LMD, LSD, HMD, and HSD), showing that two lines had a piece of comparative drought tolerant components.

Oddly, many GO terms were particularly advanced in the DRGs of the drought tolerant line H21, including six natural interaction terms (e.g., “lipid metabolic interaction”, “cell cycle”, and “formative cycle”), six atomic capability terms (e.g., “record factor action”, “record controller movement”, “kinase action”, and “transferase action”), and one cell part terms (“cytoskeleton”).

These outcomes showed that the support of root improvement, the dependability of cytoskeleton, and the propulsion of different administrative frameworks under drought might add to the drought resilience of H21. The Kyoto reference book of qualities and genomes (KEGG) improvement examination showed a sum of 10 metabolic pathways were addressed in the two lines (Figure 4).

“Plant chemical sign transduction” and “starch and sucrose digestion” were completely advanced in both of the two lines, yet the importance levels in H21 were generally higher than those in 478. Similarly, the pathway of “phenylpropanoid biosynthesis” was just fundamentally enhanced in H21 under MD and SD. These outcomes showed that these three metabolic pathways could assume an essential part in drought tolerant of H21.

### 4.3. GSEA Method

GSEA analysis is based on the expression information of all genes, sorts the genes with Signal2Noise as the standard (the default is in descending order), and analyzes whether a specific gene set (such as all genes in a pathway or all genes in a GO Term) is more ranked among all genes. Anterior or posterior (whether there is a statistically significant and consistent difference between two biological states or two groups, the calculation of Signal2Noise is generally the experimental group relative to the control group, so forward and following refer to, respectively (Figure 5).

Whether the gene set is highly expressed in the experimental group or the control group, score the pathway or term where the gene set is located, and the score is called the ES (enrichment score) value. The permutation test was performed based on the gene set, and the significant *p* value was calculated. Finally, the standardized ES value (NES value) was corrected by various tests to obtain the FDR value. Gene sets with |NES| > 1, Nymphal < 0.05, and FDR.qval < 0.25 are generally considered meaningful (when there are fewer samples per group, the replacement type is gene sets, and a stricter FDR cutoff can be used, such as 0.05). GSEA analysis was performed only on gene sets of size 15–500, as neither too large nor too small (the number of genes contained in the gene set) were of analytical significance (Figure 6).

Using the developmental time series data, finding genes that may be coregulated and/or engaged in the same biological processes is possible. Because of this, we divided the expressed genes into 30 co-expression modules, each containing genes with a similar pattern of expression (Figure 6). Based on the MapMan annotation of 16,657 expressed genes, overrepresented functional categories in each module were found. The 30 modules were split into the first, second H21, and third 478 phases based on the time of peak expression (Figure 6). The first stage, which involved significant physiological changes, occurred when seeds emerged from dormancy. The genes, which are connected to RNA processing and cell vesicle movement, were strongly expressed in dry seeds. The gene expression peaked at H21, but following 478 it rapidly decreased. The genes known as H21 are involved in RNA binding, protein synthesis and degradation, stress, the hormones ethylene and abscisic acid (ABA), and abundant storage proteins during late embryogenesis. Further, 478 has excess functions linked to stress, mitochondrial electron transport, and lipid breakdown.

### 4.4. SNP

The maize genome has a variety of SNPs. Each chromosome has an average of 525 SNPs, ranging from 327 on chromosome 10 to 926 on chromosome 1. (Figure 7). This SNPs distribution throughout the genome matched earlier studies in maize that used the same kind of marker. High-density marker genotyping, one of the primary molecular marker systems, enables the simultaneous analysis of markers that are extensively dispersed across the genome. SNP markers offer more comprehensive genome coverage than other markers, including RFLP, AFLP, and SSR. The degree of the marker’s polymorphism, which is reflected in the genetic variety among the genotypes being studied, determines how informative the marker is (Table 1).

Figure 1 shows that the 5252 SNPs employed throughout the genomes of 293 inbred lines were highly informative. PIC varied from 0.092 to 0.375, with a mean of 0.297, whereas MAF ranged from 0.051 to 0.5, with a mean of 0.284. When analyzing the genetic characteristics of Chinese maize germplasm using SNP marker data, comparable PIC range values were discovered. In their examination of CIMMYT’s tropical and temperate maize inbred lines, PIC scores between 0.25 and 0.5 show multiallelic (Figure 8).

Due to the bi-allelic nature of SNPs, where the supreme PIC value is 0.375, higher quartile PIC values, as those discovered in our study, can be regarded as being very informative. According to these standards, 65.6% of the markers used in this investigation were extremely informative (Figure 9). The average value of MAF is used to measure the level of genetic difference in the population. Generally speaking, higher MAF are recommended to increase the average allelic differentiation. A genetic distance matrix was created between all pairings of inbred line based on the 5252 polymorphic SNPs (Table 1).

The genetic distance matrix-based cluster analysis produced four unique clusters, as illustrated in Figure 2. Separating the dendrogram clusters allowed for the closest match between the number of lines grouped and the total number of lines of the formerly recognized heterotic groupings. The statistically substantial cophenetic correlation coefficient (r = 0.953; *p* = 0.0001; 10.000) permutations showed that the cluster analysis suited the underlying genetic distance matrix rather well. According to analysis, a precise dendrogram is essential for breeders to organize their genetic material. Results showed positive cophenetic values when inferences regarding genetic diversity using microsatellites (Figure 10).

In assumption, 10 (3.4 percent) of the corn inbred lines examined in this study were categorized differently from the prior heterotic group categorization. This demonstrates the significance of combining traditional breeding with molecular breeding for a genetic diversity study that is more precise (Table 1). The categorizing of the contradictions in marker-based genotypes may be inaccurate owing to pedigree information mistakes or genetic drift during the inbreeding process. (Figure 11).

### 4.5. Correlations Analysis

It was found that each plant had a respectable amount of root, with the genotype H21 having the greatest number by around 30%. The specific number of productions measured capacity roots was 1.10 with genotype H21 having the most significant number of 5% production estimated roots. The most elevated new root yield was kept in H21 while most elevated dry root yield was kept in 478. Recorded dry matter constituent went from 23.57% in 478 to 59.45% in H21. Dry root yield was intensely and primarily connected with the number of production potential roots, the number of roots per plant, the number of roots harvested, the weight of the roots, and the number of new roots produced.

New root yield was fundamentally related to roots per plant, production roots and collect list. Protein content was deeply corresponded with gather file and dry matter substance, and adversely associated with a number of productions measured roots, roots per plant, and dry matter substance. The general commitment of the different qualities to the genotype execution was made sense of by guideline part examination, Figure 12.

In view of the PC1 coefficients, four factors were significantly dedicated to variety. PC2 made sense of 20.50% of the absolute type, with a significant commitment from the collect list, dry matter substance, and protein content. PC3 made logic of 14.03% of the absolute variety with a significant commitment from production roots, collecting wet and dry matter substance. A careful analysis of the changes that took place in the preliminary spots north of CIAT and Quilichao over the course of two years revealed that genotype was particularly crucial for the number of roots produced per plant, the weight of the roots, the number of new roots produced, the number of dry roots produced, the protein content, and the dry matter content of the properties that were evaluated (Figure 12).

The joined examination of the difference in the two areas of CIAT preliminary locales is introduced in Figure 12. There were very major variances in the direct effects of the genotype on root per plant, new root production, dry root produce, and dry matter content. Connection among genotype and area was critical for just dry matter substance. After the adjustments were made to the production and the protein content, every one of the qualities showed a decent coefficient of assurance (R2) of 0.99 across the three conditions. There was a profoundly critical area impact for every one of the characteristics.

## 5. Discussion

The current study has resulted in the development of a maize freak, H21 by 478, that is resistant to drought. In comparison to its wild type 478, it showed a higher level of drought resistance when it was subjected to drought conditions. Between H21 and 478, there continued to be significant differences in the analytical files (Figure 1). The relative water content of the leaf, often known as RWC, is a metric that is commonly used to identify the drought resistance of several agricultural crops, including corn. Under drought conditions, H21 showed a higher RWC in its leaves than 478 did; for example, it had a higher ability to maintain high moisture content. The amino acid proline and the sugar that is easily dissolved both play important roles in the osmotic guideline, which is connected with drought tolerance (Figure 2). When exposed to drought, the leaves of the H21 variety contain much more proline and soluble carbohydrates than those of the 478 variety. When maize is subjected to drought stress during the transformation from somatic to reproductive development, the plant shows a considerable reduction in grain output. In order to create drought-resistant varieties, it is essential to have a comprehensive understanding of the biochemical and gene regulatory networks involved in corn’s tolerance to drought conditions during these various stages of development. This will allow for the development of cultivars that are more resistant to drought. Even with recent advances in molecular biology tools, the complex adaptive mechanisms that underlie water deficiency stress resistance from cellular proliferation to vegetative plant growth continue to be an enigma [17].

A better productivity of the antioxidative safeguard framework might safeguard photosynthetic colors, proteins, and genetic material from abundant ROS loss. Feline action is conversely relative to MDA content under serious drought tolerance. The examiner of CAT action and MDA content demonstrated that H21 experienced a smaller amount of ROS harm than 478 under drought. Light dispersion in plant shade can be improved in the field by diminishing H21 content (Figure 3). Then again, nitrogen and energy saved by a fading H21 combination would upgrade the particular reactions to drought. In this manner, low H21 content in the leaves of H21 may be a piece of its drought variation methodology [18]. More significant, the levels of RWC, osmolyte collection, cell reinforcement exercises, photosynthetic effectiveness, and a lower degree of MDA added to the drought tolerance of H21 than 478 (Figure 4).

Evaluating the transcriptome state and mining genes associated to drought tolerance in maize may both be accomplished successfully with the use of genome-wide gene expression profiling, performed by RNA-Seq [5]. Numerous genes that are sensitive to drought or that exhibit genotypically varied expression across the two lines with divergent resistance to dry conditions were found in our investigation [11]. The opposite transcriptional regulatory predisposition that reacted to water stress in maize and the diversified gene function enhancement between drought resistant and susceptible lines providing a picture of the regulatory network in maize’s transcriptome that was linked to water stress. This was shown by the fact that there was a noteworthy variance among the two types of lines. The overrepresentation of transcriptional regulators is common in most cases, particularly sensitive genes and genotypically differently expressed genes, in the tolerant line revealed a varied mechanism for maize drought tolerance [6].

Solvent sugars such as sucrose, galactose, maltotriose, and oligosaccharides, are the types of sugars that accept dynamic fractions in osmotic adjustment when there is a drought. In drought-tolerant wheat, the expanded delivery of dissoluble sugar blend-related qualities made sense of its drought tolerance. In the current research, the solvent sugar content in H21 was greater than that in 478 under the influence and drought disorders (Figure 5). Obviously, H21 has an effective drought transformation system mostly founded on the osmoregulation of dissolvable sugar [5]. In the present research, the discharge of photosynthesis-related qualities in H21 was practically unaffected, hence H21 could compose solvent sugar all the more effectively under drought conditions. In order to identify important regulatory genes and gene co-expression pathways that are involved in the maize plant’s response to environmental stress, we used RNA-seq based technique to undertake a thorough relative transcriptome study of drought-resistant lines from the somatic to propagative development stages. In order to support the RNA-seq data, we also measured photosynthetic parameters. In addition, a functional validation study using qRT-PCR provided evidence for the differential expression of the genes that were discovered. Our insights not only improve our knowledge of the processes that enable maize to survive the effects of drought stress, but they also provide a genetic resource that is both efficient and cost-effective for the genetic improvement of corn (Figure 6).

At the level of the transcriptome, H21 guided the activity of a specific component of the sugar solution. Glucose has the potential to form trehalose through a 1,1-glycosidic bond, so mitigating the damage produced by high contents of trehalose-6-phosphate under conditions of prolonged drought. In the current study, phosphatase’s behavior was restrained in 478, while their expression displayed a sharp fluctuation under drought (log 2 overlay changes from 3.46 to 7.63), indicating that the increased substance of solvent sugars in H21 probably will not be expected to the improved trehalose combination (Figure 7). Additionally, both 478 and H21’s -amylase outputs were activated by dryness, indicating that the leaves’ starch could be broken down into maltose [19].

Using two breeding lines of corn that had varied levels of tolerance to droughts, researchers used RNA-Seq technology to investigate an amount of DNA that were susceptible to drought, as well as proteins that showed differential expression across variants [2]. Between the resistant and susceptible lines, researchers discovered significant differences in the regulatory patterns and operational abundance of these proteins. The fact that drought resistance and resilience were both enhanced by the overrepresentation of certain transcription factors in the tolerant line [18] demonstrates how important these variables are. Our findings subsidize to an improved considerate of the molecular processes involved in drought response and to the identification of candidate genes for future research in the quest to improve maize’s resistance to drought [14].

In summary, H21 was able to accumulate more solvent sugars in leaves and keep up with the consistence of photosynthesis during dry conditions, which resulted in it being more drought resistant than 478. The contribution of characteristics associated to the biosynthesis of cell walls to the preservation of water in H21. The incorporation of a synchronized multienzyme complex in the process of polysaccharide production provided regions of stability for the combination of a cell wall (Figure 8). The increased supply of glycosyltransferase contributed to the drought-induced cytokinin deficiency in *Arabidopsis thaliana* freaks. The investigation of GO development revealed that in H21 a number of DEGs were associated with cell wall connection. This was notably the case with the development family, which was induced by a variety of abiotic stressors and ABA. It was observed that increasing stomatal expression levels decreased their thickness (Figure 9).

Moreover, expansions could decrease water misfortune by ceasing cell wall movement and hardening cell structures [15]. There was a strong correlation between drought resistance in maize and both the lignin concentration and the middle outcome of the phenylpropanoid mix route (Figure 11). The accumulation of caffeic abrasive and p-coumaric abrasive in the xylem sap influenced the growth of maize leaves during periods of drought by having an effect on lignin production. The abrasive effects of p-coumaric acid might rapidly lignify tissues when placed under strain and further enhance adaptability when exposed to drought. In the most recent study, KEGG and GO enhancement studies revealed that lignin-related metabolic reactions, such as the phenylpropanoid blend route, were improved in H21 under drought conditions. This was shown to be the case in the plant. The most likely scenario is that the phenylpropanoid production pathway and its components, including caffeic corrosive and p-coumaric corrosive, were accountable for reaction of H21 to drought (Figure 12).

Plants’ ability to withstand drought is a complicated process with varied responses. At the molecular scale, as well as the cellular scale, a wide array of proteins are engaged in the process of adapting to and responding to drought. The genes that code for functional proteins and the genes that regulate those proteins may be used to classify these genetic traits [9]. There are many different types of structural proteins, some of which include transporters, detoxifying enzymes, osmolyte biosynthesis enzymes, late embryogenesis abundant (LEA) proteins, and others [15]. In order to regulate transcription factors, a regulatory network configuration that incorporates stress detection, cell signaling, and functional gene regulation has been designed [16].

A major objective of the study was to demonstrate that classifying maize lines using high-density molecular marker data is a precise approach that does not require extensive field-combining ability studies [8]. The current study’s findings are consistent with past genetic diversity studies that have found SNPs to be a valuable tool for inbred grouping lines into genetically related groups and for guiding hybrid crossings between members of various groups to provide superior yield performance. SNPs are also helpful for genotyping and integrating new inbred lines in a new cluster analysis, which can help to breed by swiftly detecting the genetic link of new inbred lines. Trace endogenous substances come in many forms as plant hormones. The loss in plant growth under drought-stress circumstances may exist due to an altered hormonal balance. Ultra-trace levels of plant hormones might play a crucial role in plant development, growth, and fast response to abiotic and biotic stressors. The Brassinosteroid C-6 Oxidase, which catalyzes the last steps of brassinosteroid production, is encoded by the Brassinosteroid-deficient dwarf1 gene. It is also interestingly linked to documented height loci in maize. Plants that overexpress ABA, which is necessary for drought resistance, show improved drought tolerance. However, ABA 8′-hydroxylase hydroxylates ABA to 8′-hydroxy- abscisate and NADP^+^, depleting ABA to lower ABA levels and resulting in phenotypes ABA-deficient.

Comparing the categorization of maize lines conducted earlier using pedigree and breeding data to the classification achieved using SNPs, revealed good accuracy. Although the majority of the genotypes in this research fell into one of the four heterotic categories that the company’s breeding program had previously developed, there were be seen, demonstrating the large amount of genetic diversity present in maize germplasm [9]. Furthermore, the genotyping information derived from this work may be used to further hybrid prediction models, allowing for more precise identification of hybrids, and increasing breeding effectiveness [12].

## 6. Conclusions

We can state with confidence that the results of our in-depth comparative study of the gene enrichment analysis of two distinct maize hybrids to drought stress at the 478-development stage are presented here. Genetically, in comparison to the sensitive line 478, the drought-resistant line H21 had a greater number of enzymes, lower levels of oxidative stress, and improved cell water holding capacity. Overall, the genome our RNA-seq data identified many differentially expressed genes as being asserted in response to drought, with several of these genes being particularly disclosed in H21. We found that the expression of genes related to earlier identified pathways involved in the drought stress response, such as those involved in the manufacture of secondary metabolites, TF regulation, detoxification, and stress resistance, increased in response to drought stress. Our results provide a foundation for future research utilizing targeted cloning and aid in our thoughtful investigation of the mechanisms governing maize’s drought resilience.

## Figures and Tables

**Figure 1 genes-14-00031-f001:**
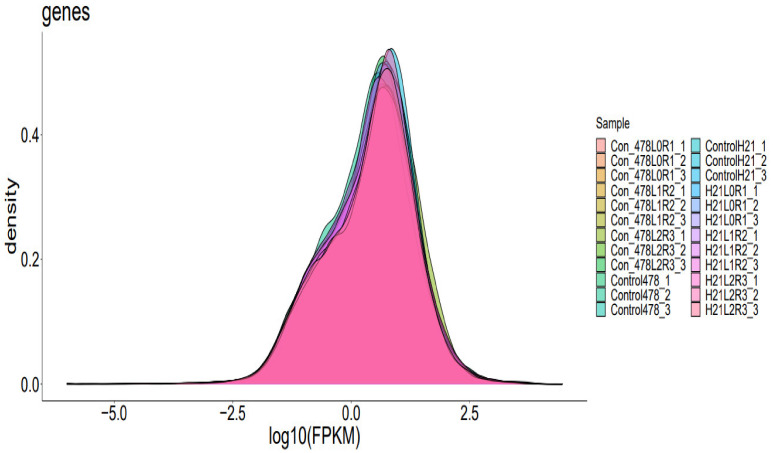
Visual representation of genes expression density of maize, showing a sharp peak at high density.

**Figure 2 genes-14-00031-f002:**
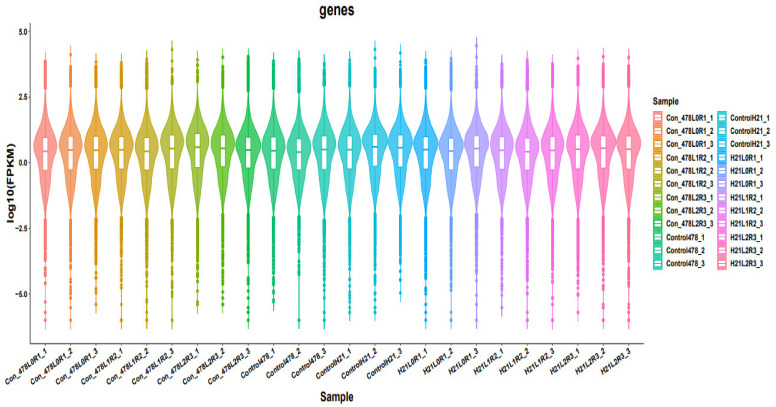
Different samples are analyzed simultaneously and are presented in the form of violin.

**Figure 3 genes-14-00031-f003:**
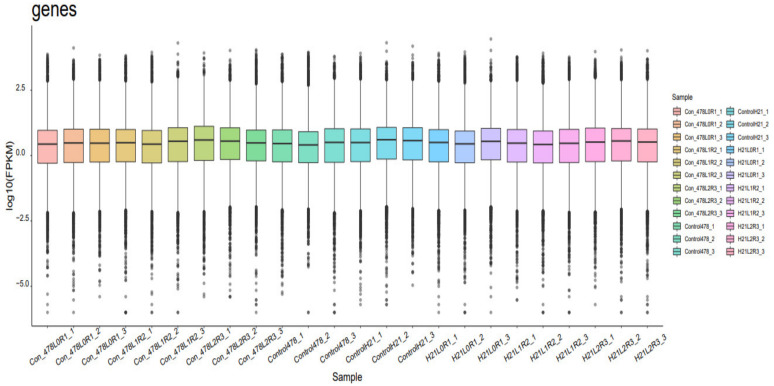
Different samples portraying GO analysis and each sample is mentioned as individual gene.

**Figure 4 genes-14-00031-f004:**
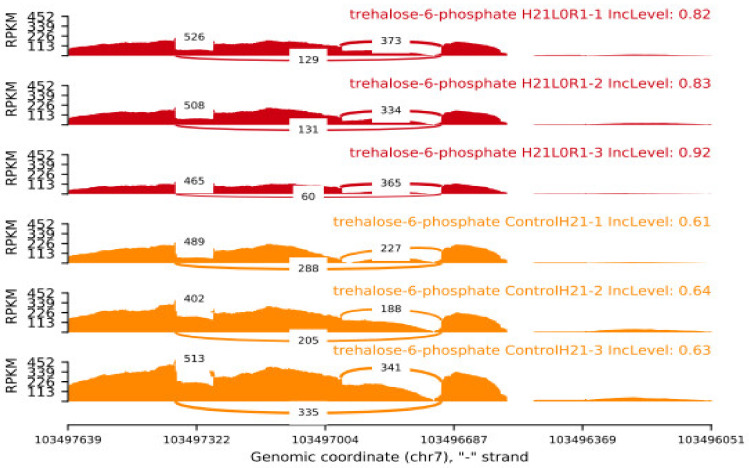
GO analysis different peaks are shown clearly in the figure with different sample sizes.

**Figure 5 genes-14-00031-f005:**
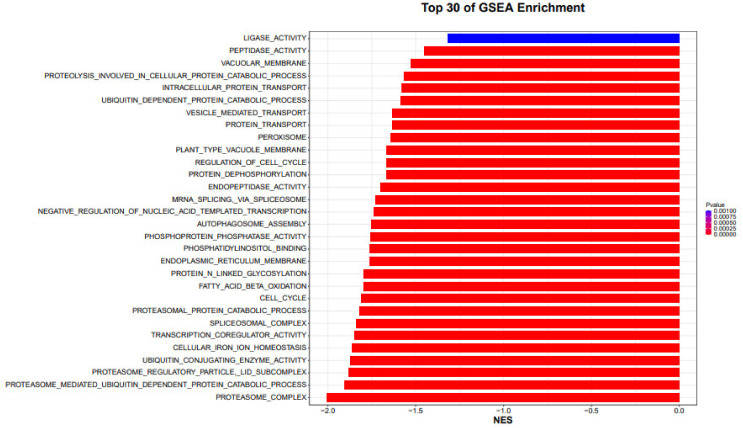
GSEA analysis top 30 samples of GSEA enrichment analysis are shown in the form of a horizontal bar graph.

**Figure 6 genes-14-00031-f006:**
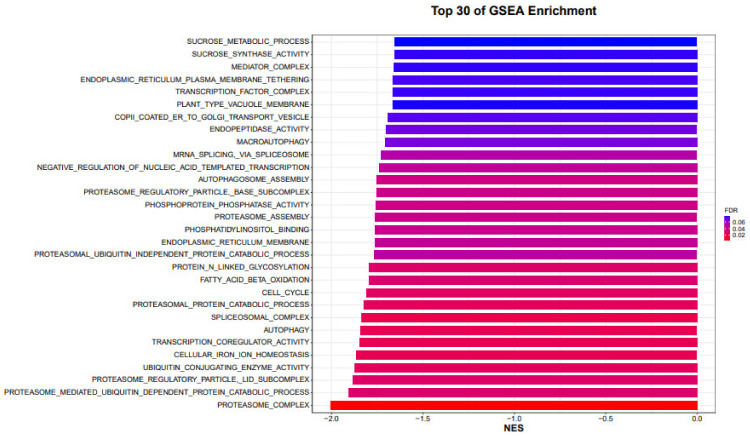
GSEA analysis top 30 samples of GSEA enrichment analysis are shown in the form of a horizontal bar graph.

**Figure 7 genes-14-00031-f007:**
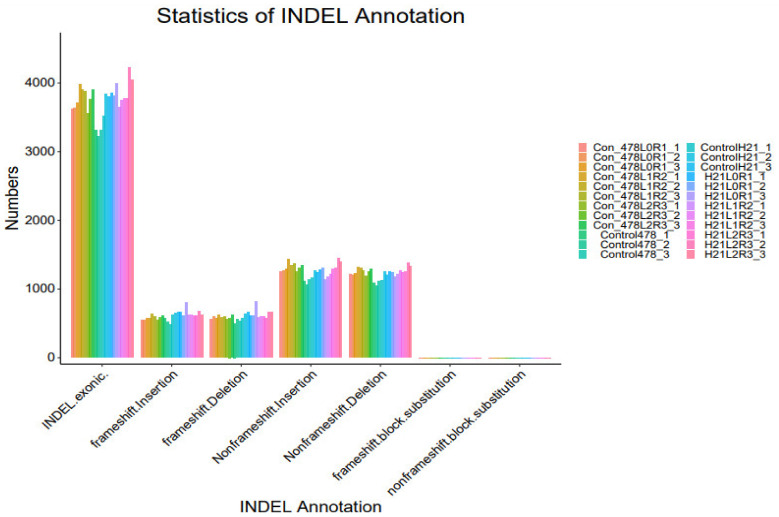
SNP analysis results INDEL analysis are shown in the form of a bar graph.

**Figure 8 genes-14-00031-f008:**
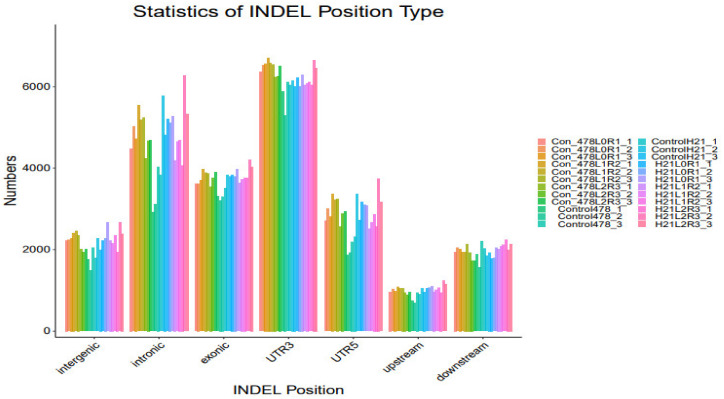
SNP analysis results INDEL analysis are shown in the form of a bar graph.

**Figure 9 genes-14-00031-f009:**
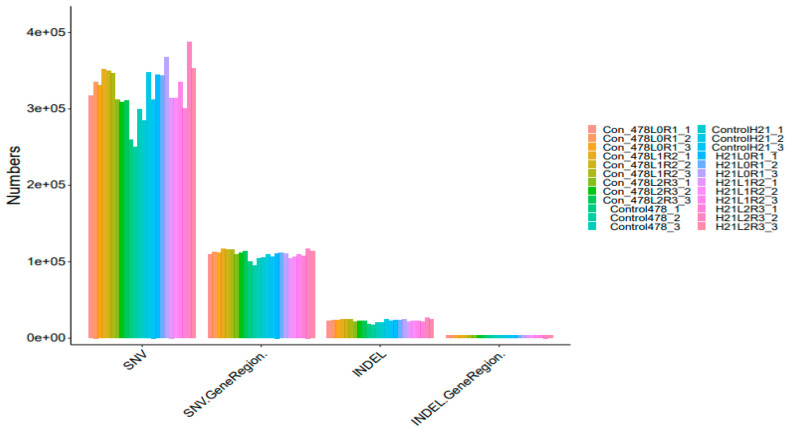
SNP analysis results INDEL analysis are shown in the form of a bar graph.

**Figure 10 genes-14-00031-f010:**
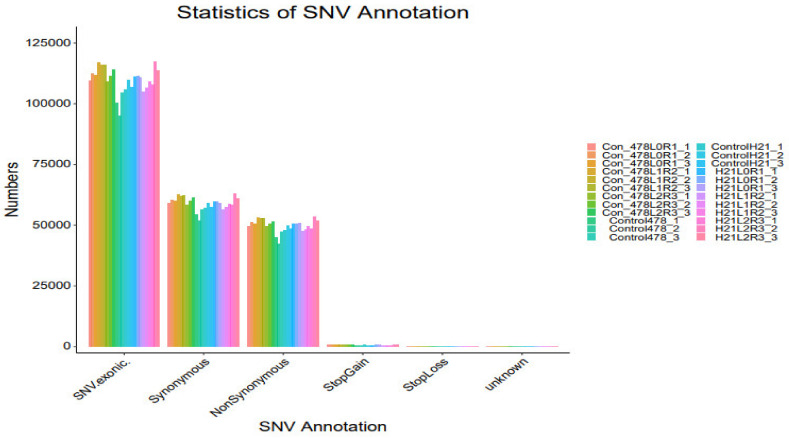
SNP analysis results SNV analysis are revealed in the form of a bar graph.

**Figure 11 genes-14-00031-f011:**
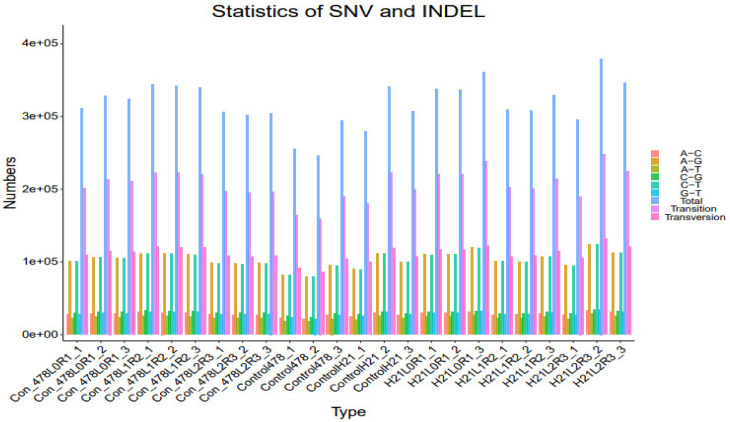
SNP analysis results SNV analysis are shown in the form of a bar graph.

**Figure 12 genes-14-00031-f012:**
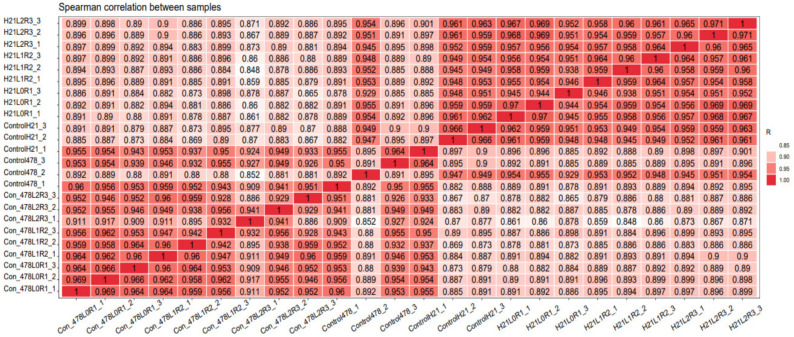
Correlation cluster of maize Spearman correlation of different samples that showed a comparative analysis.

**Table 1 genes-14-00031-t001:** Results of SNP analysis.

Sample	SNV	SNV (Gene Region)	INDEL	INDEL (Gene Region)
Con_478L0R1_1	317,447	109,631	22,372	3621
Con_478L0R1_2	335,578	112,446	23,550	3638
Con_478L0R1_3	330,907	111,822	23,177	3711
Con_478L1R2_1	351,518	117,124	25,118	3987
Con_478L1R2_2	349,311	116,003	24,412	3906
Con_478L1R2_3	346,695	116,230	24,528	3875
Con_478L2R3_1	311,727	109,236	21,552	3560
Con_478L2R3_2	308,668	111,474	22,214	3766
Con_478L2R3_3	310,894	114,079	22,806	3908
Control478_1	259,539	100,510	18,424	3318
Control478_2	249,938	95,025	17,361	3226
Control478_3	299,808	104,763	20,935	3316
ControlH21_1	284,771	105,851	20,496	3518
ControlH21_2	347,813	109,906	24,376	3841
ControlH21_3	312,458	106,749	22,326	3805
H21L0R1_1	344,214	111,197	23,577	3847
H21L0R1_2	343,231	111,465	23,266	3817
H21L0R1_3	367,762	110,892	24,537	3990
H21L1R2_1	314,609	104,783	21,679	3643
H21L1R2_2	314,381	106,501	22,467	3746
H21L1R2_3	335,042	109,192	23,055	3781
H21L2R3_1	300,528	107,825	21,661	3775
H21L2R3_2	387,144	117,458	26,906	4221
H21L2R3_3	353,192	113,905	24,772	4048

## Data Availability

Not applicable.

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
