# Peer review of "Comparative Gene Enrichment Analysis for Drought Tolerance in Contrasting Maize Genotype"

_genes, 2022, doi:10.3390/genes14010031_

Round 1
Reviewer 1 Report
The title „Comparative Gene Enrichment Analysis for Drought Tolerance 2 in Contrasting Maize Genotype ” is suggestive and suitable for the content of this work.
The present ms. is a good piece of work presenting valuable scientific data. These kinds of studies are needful for a sustainable agriculture.
In the abstract, the justification for the research (its importance) should be specified more clearly and should be included in the abstract.
The rest of the work, introduction, methods and results are complex and well organized, fact that shows the correct and detailed documentation of the authors,
Please provide reviews in support and contradiction of your results. There should be at least 10-15 relevant reviews, in order to compare the results with other researchers. Please also explain how the results relate to previous findings, whether in support, contradiction, or simply as added data.
I also recommend a short linguistic check.
The conclusions are thoroughly supported by the results presented in the paper. Congratulations for the work!
Author Response
Reviewer 1
The title „Comparative Gene Enrichment Analysis for Drought Tolerance 2 in Contrasting Maize Genotype” is suggestive and suitable for the content of this work.
R: Thanks for your comments and valuable suggestions.
The present ms. is a good piece of work presenting valuable scientific data. These kinds of studies are needed for sustainable agriculture.
R: We are thankful for your recommendation of this work.
In the abstract, the justification for the research (its importance) should be specified more clearly and should be included in the abstract.
R: Thanks for the comments. We have added the importance of the study in the abstract section as we mentioned in the conclusion section.
The rest of the work, introduction, methods, and results are complex and well organized, the fact that shows the correct and detailed documentation of the authors
R: Thanks for the comments.
Please provide reviews in support and contradiction of your results. There should be at least 10-15 relevant reviews, in order to compare the results with other researchers. Please also explain how the results relate to previous findings, whether in support, contradiction, or simply as added data.
R: Thanks for the comments. I have added some reviews as you suggested. The comparison of the current study with previous studies is given in the discussion.
I also recommend a short linguistic check.
R: Thanks for the comments. I have checked all manuscripts and improved the language of manuscript.
The conclusions are thoroughly supported by the results presented in the paper. Congratulations on the work!
R: Thanks for your comments. We hope to improve our work in the future.

Reviewer 2 Report
Quality of figures can be improved.
Add picture of maize plants at the time of harvesting.
Add heat map of gene expression obtained from qPCR.
Add about how results of RNA Seq data were verified?
Author Response
Reviewer 2
The quality of the figure can be improved.
R: Thanks for the comments. I have improved the quality of the figures. These Figures are made with standard procedures.
Add heat map of gene expression obtained from qPCR.
R: Thanks for the comments. Heat map of gene expression obtained from qPCR already published. Paper references can be provided if you need them.
Add about how results of RNA Seq. Data were verified?
R: Thanks for the comments. I have mentioned this in the manuscript as the reviewer suggested.
Please add the picture of maize at the time of harvesting
R: Dear reviewer, at the moment we do have not a good-quality picture of this stage (harvesting).

Round 2
Reviewer 2 Report
Authors have addressed all the comments. It can be accepted for publication.